# The Communicative Behavior of Russian Cosmonauts: "Content" Space Experiment Result Generalization

**Dmitry Shved \*** , **Natalia Supolkina and Anna Yusupova**

State Scientific Center of the Russian Federation—Institute of Biomedical Problems, Russian Academy of Sciences, Khoroshevskoye Hwy, 76A, 123007 Moscow, Russia; natalyasupolkina@yandex.ru (N.S.)

\* Correspondence: d.shved84@gmail.com

**Abstract:** The increasing complexity of the space flight program and the increase in the duration of missions require an improvement in psychological monitoring tools for astronauts in orbit. This article summarizes the experience of using quantitative content analysis of communication between the crews and the Mission Control Center (MCC). This method allows us to assess the dynamics of an astronaut's psycho-emotional state, identify their communicative style, and detect the communication phenomena of board-MCC communication. The method is based on a combination of the coping strategies approach by Lazarus and Folkman as well as B.F. Lomov's concept about the three functions of communication. We found the influence of workload on the structure and volume of communication, defined the main stable communication styles of crewmembers, and confirmed the presence of the emotional transfer phenomenon. We detected that astronauts successfully solve problems that arise in orbit using the capabilities of their communication style. An ineffective MCC communication style usually leads to psycho-emotional ill-being, manifesting in the emotional transfer phenomenon. The presence of the "third-quarter" phenomenon was not confirmed by materials from six-month space flights.

**Keywords:** crew communication; content analysis; coping strategies; communication styles; emotional transfer; third-quarter phenomenon

## 1. Introduction

The issues of studying communication peculiarities between Russian cosmonauts (we shall further refer to as astronauts) and the Mission Control Center (MCC) become even more crucial as ultra-long-term (approaching two years for possible flights to Mars) flights, including interplanetary flights, are becoming a close perspective. We aim to show these communication peculiarities in this paper, where we summarize the results of a three-year "Content" space experiment, where we developed a methodology for content analysis [1], which is a systematic, reproducible method of reducing an array of text into a limited number of categories. We used this method to analyze transcripts of daily communication between crew and the MCC, collected a statistically reliable data corpus, and highlighted significant features of communicative behavior during a long-term space flight as well. The data on communication phenomena in long-term space flights that are obtained in the "Content" experiment make it possible to assess the risks for the implementation of the work schedule that are posed by conflicts between crews and the MCC, as well as possible technical failures in the crew–Earth communication circuit during interplanetary flights. In addition, the research results led us to the conclusion that in order to maintain efficient data transmission and maintain the optimal performance of astronauts in upcoming ultra-long-term flights, it is necessary to think about significant changes in the practice of organizing communication between crews and the MCC. These changes should include reducing control and direct management from supervisors and greater independence in decision making for astronauts.

In 2015–2018, the Institute of Biomedical Problems of the Russian Academy of Sciences, together with RSC Energia, held the "Content" experiment that was aimed at studying the content of open-channel communication between Russian crewmembers and Russian MCC during long-term space flights on the International Space Station (ISS).

The purpose of the experiment was to carry out a quantitative content analysis of the professional communication of astronauts with MCC for the rapid assessment of their psychophysiological state, as well as intra- and intergroup (crew–MCC) interactions. The obtained data were supposed to be used for the early diagnosis of astronauts' mental well-being with the aim of providing subsequent psychological support measures. We were also searching for evidence of a so-called third-quarter phenomenon (an increase in stress and emotional problems experienced by people working in confined habitats after the halfway point of their mission) in space flight.

The main objectives of the space experiment were as follows:

1. To approbate the method (previously used in ground spaceflight simulations) of psychophysiological monitoring based on an analysis of the crew's communications with the Control Center and to refine the list of content analysis criteria according to the real space flight communication practice.

2. To assess the dynamics of the psycho-emotional state of astronauts during a long-term flight according to an analysis of their communication with the MCC and to identify the influence of workload levels, critical flight periods, and significant events on the structure of crew communication.

3. To study personality-based, sustainable communication styles of astronauts.

4. To study communication problems between astronauts and the Earth (MCC mission controllers and specialists).

When developing the methodology for analyzing astronauts' communication, we proceeded from the premise that despite the subjective control over their conversations with the MCC, which, as the astronauts know, are recorded and transmitted through several communication channels (including the Internet), they communicate quite freely—and therefore we can identify significant diagnostic information about their psycho-emotional state using speech analysis.

Quantitative content analysis [2,3] was used to analyze the astronauts' speech. The unit of communication analysis that we used is a statement expressing a complete thought [4].

The system of content analysis categories that we use was developed on the basis of R. Lazarus and S. Folkman's stress-coping approach [5] and its application to astronauts' speech content analysis by P. Suedfeld [6,7]. These categories are arranged in accordance with B. Lomov's theory about the main functions of communication in professional performance [8]. Describing stress-coping strategies, R. Lazarus and S. Folkman point to the wide range of resources that people utilize to cope, including external ones, e.g., instrumental and social support, and internal ones, e.g., self-regulation, motivation, and social and professional skills. These strategies target problem solving or emotional regulation under stress. P. Suedfeld, who analyzed the content of astronauts' diaries and interviews, confirmed that participation in space flight requires the utilization of coping strategies in order to withstand stress caused by a deficit of instrumental and social resources [6,7,9].

According to B. Lomov, the communication of human operators implements three main functions: (1) informing or data exchange; (2) social regulation and social roles distribution (subordination); and (3) affective function related to the expression of emotions. We support the author's idea that in professional communication, i.e., in the crew-MCC talks, mutual informing, exchange of data, planning, initiative, and recommendations should dominate expressions of social regulation and emotions. According to our initial hypothesis, later confirmed via the obtained results [10], an increase in the number of statements aimed at social interaction in the crew talks with the MCC, combined with an increase in emotional statements (positive or negative), indicate rising levels of psychological stress. Depending on these theoretical approaches and data from space simulations, we defined

semantic indicators that allowed experts to attribute statements to the communicative functions (informing, social regulation, or affective) that they execute in the talks.

Thus, starting in 2000 using the Bales method [11] and the ideas of Lazarus and Lomov and consistently modifying them during long-term isolation experiments (SFINCSS'99, Mars 105, Mars 500: [12]), the team of authors approached the beginning of the "Content" experiment with a methodology [1] that required clarification of the content analysis method based on the specific features of work activities in space. For this purpose, a pilot study onboard the ISS was conducted with the participation of American colleagues, which allowed us to test the validity of our method [13].

Since another objective of the study was to study the personality-based, stable communication styles of astronauts, we needed to develop an appropriate methodological approach. We based the classification of communication parameters within communication styles upon V. Satir's concept. She observed that people tend to react to stress and threats to their self-esteem (endangering one's perceived independence and professionalism) with one of four different defensive communication styles [14,15]. The perceived lack of trust and rejection of their position by Earth specialists were described as the main problems of communication with the MCC by astronauts and astronauts earlier in post-flight interviews [16,17]. The Satir model corresponds to B.F. Lomov's concept of the three functions of communication, which we rely on in the analysis. Satir considered the stylistic features of communication in the light of information exchange, i.e., how much a certain style helps to solve problems, improves or complicates the transmission of information (the communicative function according to B.F. Lomov [8]), or, on the contrary, replaces effective interaction by discussing relationships (B.F. Lomov's function of social regulation) and experienced emotions (affective function).

V. Satir identified five styles of communication in a closed loop of communication. We proceeded from the fact that those who use the distractor style are preliminarily screened out during psychological selection procedures for the astronauts' corps. The leveling style, in our opinion, would be a desired standard of space communication, but would not be useful for communicative behavior in an extreme situation. Therefore, three main styles were sought after and analyzed—blaming, computing, and placating.

A blamer is critical, complaining, and a faultfinder, angry because they anticipate not getting their needs met. Their learned defense for this is to go on the offensive. Blamer behavior finds fault while having trouble accepting responsibility. Blamers are more likely to initiate conflict. Placaters are non-assertive, never disagreeing and seek approval. They avoid conflict. Their main concern is how other people perceive them. The computer (super reasonable) is cool, calm, and collected but displays no emotion, masking a feeling of vulnerability. They expect people to perform efficiently and conform to the rules.

For our study, we chose V. Satir's classical communication model (styles of communication, 1972) for three main reasons. Firstly, the model identifies the main communicative characteristics of a person in a stressful situation—and we study the communicative behavior of astronauts under the influence of stress factors of long-term space flight. Secondly, V. Satir's model corresponds to B.F. Lomov's concept of the three functions of communication, which we rely on in the analysis. V. Satir examined the stylistic features of communication via information exchange, i.e., how does a style help to solve problems, improve or complicate the flow of information (this corresponds to the communicative function concept according to B.F. Lomov [8]), or, on the contrary, replace and affect interaction with discussion on relationships (social regulation function) and experienced emotions (affective function). Finally, taking into account the prospects for using the methodology in standard MCC practice, V. Satir's classification is attractive for its simplicity and practicality: it is easy to understand, remember, recognize, and apply. As a practicing psychotherapist, V. Satir built this model to diagnose communication patterns in families associated with the experience of stress and requiring correction.

## 2. Materials and Methods

### 2.1. Participants

The subjects were male Russian astronauts of ISS 43/44–54/55 flights who took part in the "Content" space experiment; N = 14, age range 40–57. Among these astronauts, 7 subjects had an experience of 1 or 2 flights (including the ones incorporated in our studies), and 7 subjects made 3 to 6 flights.

### 2.2. Bioethics and Informed Consent

The studies involving human participants were reviewed and approved by the Bioethical Commission of the Institute of Biomedical Problems of the Russian Academy of Sciences and fully complied with the principles of the 1964 Declaration of Helsinki.

Each study participant voluntarily signed an informed consent after having the potential risks, benefits, and nature of the upcoming study explained to them.

### 2.3. Design of the Study

The studies were conducted within the frame of "Content" space experiments involving Russian ISS crewmembers. The experiment was dedicated to the psycholinguistic analysis of crew-MCC communication and was aimed at searching for indicators of the psychological state and well-being of astronauts.

We studied daily crew-MCC communications during 15 ISS missions with durations from 116 to 340 days (mean 179, median 174).

A corpus of 164658 statements containing categories of interest were selected from official Roscosmos transcriptions made daily for open (non-confidential) communication channels.

### 2.4. Content Analysis Criteria

In our research, we added some strategies proposed by Suedfeld et al. (e.g., Endurance/Obedience and Humor) [6,7,9] to Lazarus and Folkman's list of copings [5]. As a result of our further studies, seven additional operational categories related to inflight data exchange (*Informing, Problem, Initiative, Effort, Claim/Complaint, Positive/negative emotions,* and *Trust/Mistrust*) were added by Russian MCC experts in order to target professional communication during problem solving more precisely.

Furthermore, independent experts divided the whole corpus of astronauts' statements in accordance with the expressed need for information exchange for problem solving and stress coping. By effective communication, we mean statements where the evident need for information exchange is expressed when the subject intends to use it for active resolution of the existing problem causing stress. By maladaptive statements (strategies), we mean those in which the subject is trying to avoid contact or open information exchange, as well as responsibility for problem resolution. Ambivalent statements (that cannot be defined as adaptive or maladaptive) do not contain coping expressions.

In order to neutralize the influence of communication quantity (the subject's "talkativeness") on the results of content analysis, the unit of reference is not the number of words spoken, but the statement—a fully expressed idea (explained in [4]). Based on this, the statement can consist of several words and of several sentences. Thus, the final set of 25 categories that we used to analyze communication includes not only coping strategies but also categories reflecting the functions of communication and the specifics of communication between astronauts and MCC specialists (Table 1).

To interpret the content analysis data, we also used the weekly psychological reports of the MCC and the post-flight interview data. The content analysis data were compared with the results of the weekly psychological conclusions of the MCC psychological monitoring group.

According to V. Satir's description [14,15], we highlighted communication attitudes and coping strategies that might manifest in communication in each group. We asked experienced experts from Russian MCC (four psychologists), not involved in our content

analysis experiment, but for years participating in astronauts' inflight monitoring, to become acquainted with Satir's communication model. Afterward, they were asked to classify astronauts who participated in the "Content" experiment (N = 15) into those who mostly used blaming, placating, and computing, relying on their subjective estimation and experience. Then, the "Content" coders' group, who did not know the results of these estimations, made" blind" content analysis of the subjects' inflight talks during days with standard and intensive workloads to identify the profile of dominating coping strategies (style) for each subject. Thus, we obtained accordance between the astronauts and the communication styles: in the analyzed group of 14 astronauts, 6 were attributed with the "blaming" style, 5 with the "computing" style, and 3 with the "placating" style. A further comparison in the type of dominating coping strategies was made between the groups. In our previous studies, we showed that one of the three Satir styles ("computing", "blaming", or "placating") usually dominates in an astronaut's speech [10].

**Table 1.** Coping-based content analysis categories divided by their functions in communication (according to B. Lomov) and communication effectiveness.

| | **Communication Functions** | | |
|---|---|---|---|
| Communication effectiveness | "Informing" | "Social regulation" | "Affective" |
| Effective/Adaptive | Initiative<br>Planful problem solving | Accepting responsibility<br>Trust<br>Support | Humor (positive)<br>Self-control<br>Positive reappraisal<br>Positive emotions |
| Neutral | Informing<br>Problem<br>Effort<br>Requests/demands<br>Time<br>Cognitive load<br>Searching items<br>Equipment failure/breakdown | Seeking for social support<br>En-durance/Obedience | |
| Ineffective/Maladaptive | Escape/avoidance<br>Claim/Complaint | Confrontation<br>Mistrust<br>Responsibility avoidance<br>Self-justification | Distancing<br>Negative emotions<br>Sarcastic humor |

This content analysis method was also successfully used to study the crew-MCC communications in a series of IBMP-based model experiments (SIRIUS) [18].

### 2.5. Statistical Analysis

The data were normalized as a rate of statements per week for analysis and were processed using StatSoft Statistica 13 and Statgraphics 18SPSS software; the methods used were principal component factor analysis (Varimax rotation method with Kaiser normalization), Kruskal–Wallis H test, Wilcoxon W-test, and Mann–Whitney U-test. The nonparametric criteria were chosen due to the fact that in the normality check for all data variables (categories of content analysis), pronounced skewness (to the right) and kurtosis were detected.

To validate the proposed approach, in 2014, a pilot study was held to check the intercoder validity of the content analysis method. Four experts took part in the pilot study. They assessed a monthly data set with ISS crew–MCC communications using the 19 main content analysis categories. To assess the consistency of expert opinions, the Spearman rank correlation method was used. The opinions of each expert on 19 assessment indicators were compared with the opinions of the group (3 experts). To calculate the group's "raw" score

for each indicator, the graphical median method was used. The final agreement coefficients for each expert and group turned out to be reasonably high (rs = 0.76–0.89) to consider the technique reproducible and valid.

## 3. Results

*3.1. Assessing the Psycho-Emotional State Dynamics in Astronauts via Their Communication Analysis during a Long-Term Flight*

The Influence of Workload Levels, Critical Flight Periods, and Significant Events on the Structure of Crew Communication

To understand the impact of such flight events as problematic situations, accidents, and breakdowns on the intensity and structure of communication in flight, a three-year corpus of communications between astronauts and ground services was divided into two clusters based on our assessment of the intensity of the work schedule: communication on "quiet days" (neutral or days with a standard workload) and "problem days" (days with an increased workload).

- Days with a standard workload: these are weekdays and weekends during which the volume of planned work remained within the limits allowed by regulations and work activities which did not require any additional effort from the astronaut.
- Days with a high workload were as follows:
  1. Days of docking and undocking of manned and cargo transport ships, as well as three days before and after these events, when additional work was carried out to unload and load the ship;
  2. Days of extravehicular activity, as well as the days preceding and following the event (when equipment and spacesuits were being prepared and loaded, unloaded, spacesuits were dried, etc.);
  3. Days on which accidents and breakdowns occurred that required an immediate response from crewmembers and/or a shift in the astronaut's work schedule (for example, reducing time for meals or performing night work);
  4. Scheduled work on weekends or holidays that required more time than supposed by the norms allowed by regulations (3.5 h).

An increasing workload in flight significantly changes the volume of communication; on days with a high workload, the average number of statements in crew negotiations was 14.84, while it was 6.34 ($p < 0.05$) on days with a standard workload (Figure 1). In problematic situations, the professional crew proactively discussed possible solutions to problematic situations with the MCC.

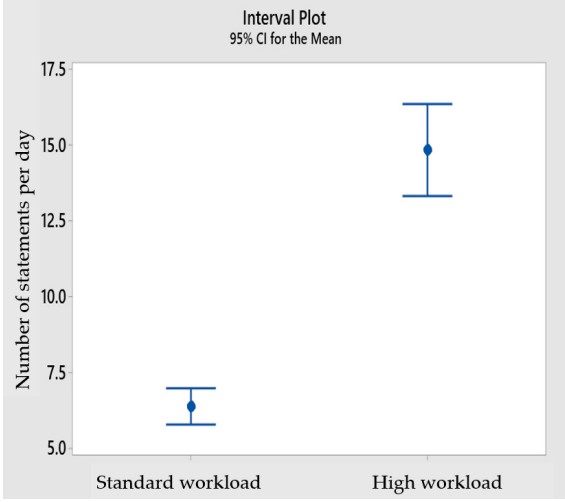

**Figure 1.** Volume of astronauts' communication (number of statements) on days with different workloads (averages per day).

It is obvious that the necessity to solve operational problems unforeseen by the flight program leads to a significant increase in the cognitive load and effort of the crew. Accordingly, the volume of communication between the crew and the Mission Control Center increases. In particular, there is an increase in statements in the categories *Problem*, *Accident/Breakdown*, and *Effort*, as well as in the *Time* category (Table 2, [19]). The growth of the latter indicator shows that quite often, problem situations are accompanied by a violation of the schedule and a shortage of working time that was allocated by planners for only the routine procedures of the flight program.

**Table 2.** The structure of astronauts' communication on days with different workloads.

| Categories Group | Categories | Days with Standard Workload | Days with High Workload | Significance of Differences According to the Mann–Whitney Test |
|---|---|---|---|---|
| | | Average | Average | |
| Maladaptive strategies | Negative emotions | 0.12 | 0.60 | <0.001 |
| | Claim/Complaint | 0.35 | 1.24 | <0.001 |
| | Confrontation | 0.13 | 0.59 | <0.001 |
| | Responsibility avoidance | 0.15 | 0.35 | <0.001 |
| | Self-justifications | 0.05 | 0.11 | 0.013 |
| Adaptive strategies | Initiative | 0.80 | 1.91 | <0.001 |
| | Positive emotions | 0.57 | 0.94 | <0.001 |
| | Planful problem solving (Planning) | 0.55 | 1.20 | <0.001 |
| | Trust | 0.04 | 0.11 | 0.001 |
| | Humor | 0.17 | 0.44 | <0.001 |
| | Informing | 1.95 | 3.94 | <0.001 |
| | Positive reappraisal | 0.01 | 0.03 | 0.008 |
| | Self-control | 0.08 | 0.17 | 0.001 |
| Neutral categories | Effort | 0.44 | 1.29 | <0.001 |
| | Requests/demands | 1.01 | 1.89 | <0.001 |
| | Time | 0.62 | 1.42 | <0.001 |
| | Cognitive load | 0.55 | 1.29 | <0.001 |
| | Problem | 0.62 | 1.37 | <0.001 |
| | Breakdown | 0.15 | 0.29 | <0.001 |
| | Searching items | 0.31 | 0.68 | <0.001 |
| | Seeking social support | 0.15 | 0.38 | <0.001 |

Crews responded to challenging situations with effective stress-coping strategies. This is evidenced by an increase in the number of statements related to strategies like *Constructive initiative* (reliability of the polynomial approximation $R^2 = 0.935$) and Responsibility acceptance (reliability of the polynomial approximation $R^2 = 0.822$). Despite stressful conditions, the crewmembers sought to optimize task performance, showed initiative, and took responsibility for its introduction.

In problematic situations, there was also a two-fold increase in the astronauts' statements indicating their *Trust* in the MCC specialists (m = 0.04 on days with standard workload and m = 0.11 on days with high workload). That is, in a stressful situation, the astronauts remained open and flexible within their communication strategy and were ready for an open dialogue—a broad exchange of information with MCC specialists in order to

resolve emerging problems. This indicates that well-trained professionals use a wide range of strategies to overcome stress (coping strategies), specifically in problematic situations. This differs from the manifestations of the "psychological closing" of the crew, identified earlier in model experiments (spaceflight simulations), e.g., Mars-500 [20].

At the same time, we cannot fail to note the negative impact of the increased workload. First of all, along with the increase in the number of informative statements, the proportion of statements with affective and social regulation functions increased in these stressful days. In particular, the frequency of *Self-Justifications* increased (from m = 0.05 on days with a standard workload to m = 0.11 with a high workload), as well as statements containing *negative emotions* (from m = 0.2 on days with a standard workload up to m = 0.6 on days with high workload).

On stressful days, two opposing strategies of communication behavior were related to the space flight experience. The first strategy was associated with *Taking responsibility* for solving problems, actively searching for ways to solve them, and then defending these solutions before the MCC. At the same time, astronauts often associate the occurrence of these problems with the shortcomings of supervisors, specialists, planners, etc. Accordingly, in the communications, along with a two-fold increase in the number of statements about *Planning* and taking the *Initiative*, there was an increase in *Confrontation* and *Claims* (Table 2). This strategy was more typical for experienced astronauts making their third (or more) flight. The second strategy was associated with *Postponing responsibility*, *Seeking* external *support*, and delegating decisions to another person (for example, an MCC specialist). This strategy showed a two-fold increase in the *Search for Support, Submission,* and *Self-Justification* (Table 2) and was mainly found in novice astronauts.

### 3.2. Influence of Flight Periods on the Crew-MCC Communication Structure

We analyzed adaptation dynamics to space flight conditions in astronauts via the volume and structure of communication content but obtained no clear results. In a medium-duration space flight (3–6 months), there was a tendency to increase the volume of communication in the first 3–4 weeks of the flight, as well as in the final 4–6 weeks. During these periods, the number of *Requests for information* increased, and there was an increased *Seek for support*. In addition, at the end of the flight, the astronauts' speech was characterized by an increased emotionality. However, the changes noted were trends and were not statistically significant.

The data obtained during the prolonged flight had significant differences. Almost all crewmembers in the second half of the prolonged flight showed an increase in the total volume of communication that concerned an increase in the work intensity, occurrence of problems of the final stage of the flight (including stowage, placement of cargo, equipment preparation for the descent, etc.). Solving these problems led to an increase in statements reflecting greater attention to workflow *Planning* (linear approximation reliability $R^2 = 0.983$) (Figure 2). At the same time, there was an increase in the number of statements related to the manifestation of *Demands/requests* toward the MCC (reliability of linear approximation $R^2 = 0.923$), the desire to *Avoid responsibility* for the problems (reliability of linear approximation $R^2 = 0.934$). Overcoming the difficult period was facilitated by the more frequent use of *Humor* by the astronauts when discussing problem situations (linear approximation reliability $R^2 = 0.947$). According to R. Plutchik, humor allows one to make a positive reassessment of the situation in order to reduce emotional stress [21]. Thus, the total number of statements containing coping strategies, both effective and ineffective, increased, which confirmed the presence of an increase in the level of psychological stress during this period.

Among the results obtained, the manifestations of the "third-quarter phenomenon" discovered during expeditions ISS-43–46 are of particular importance. Similar to the "final rush" period, during this phase of the expeditions, there was also a general increase in the use of stress-coping strategies in the crews' communication with the MCC (Figure 3). During this period, the crews that successfully completed the flight program responded to the emergence of problematic situations with effective stress-coping strategies. This is

reflected in an increase in statements related to such coping strategies as *Initiative* (reliability of the polynomial approximation $R^2 = 0.935$) and *Responsibility acceptance* (reliability of the polynomial approximation $R^2 = 0.822$). In other words, despite stressful conditions, crewmembers looked for ways to optimize task performance, showed initiative, and took responsibility for its fulfillment.

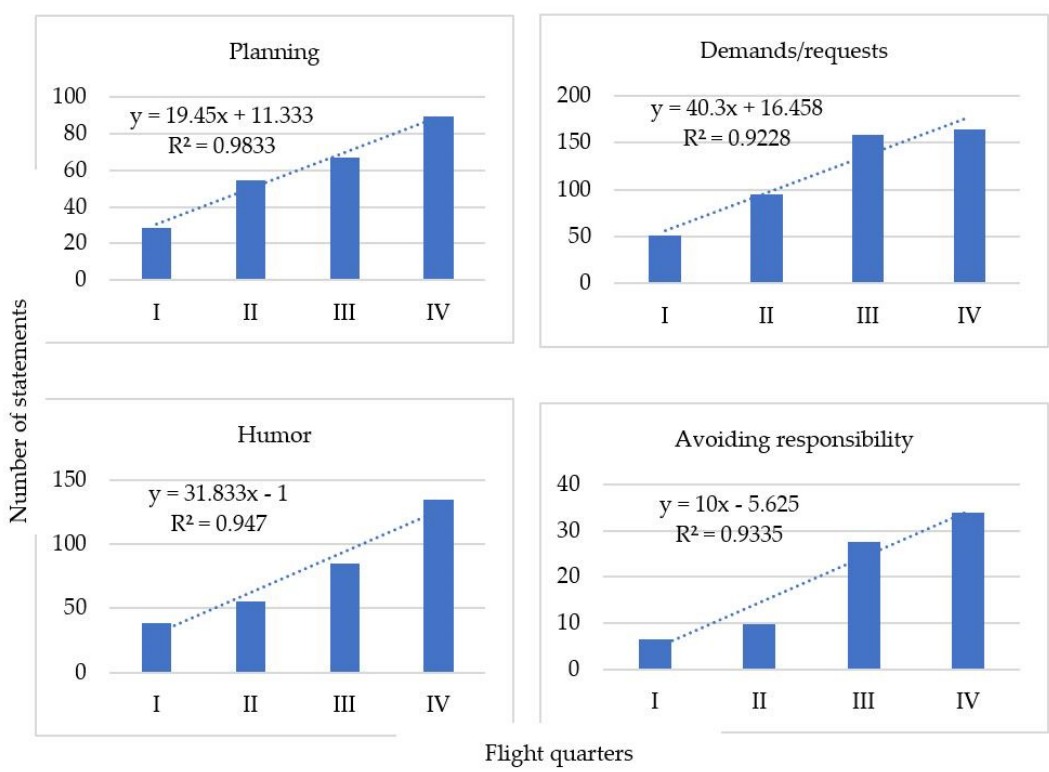

**Figure 2.** Manifestation of the "final effort" phenomenon in crew communication.

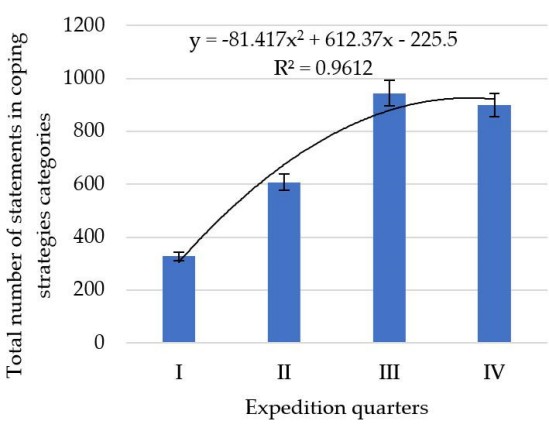

**Figure 3.** Coping strategies in crewmembers' communication during prolonged space flight.

Of particular interest is the increase in the number of statements emphasizing the importance of *Time* use by the crew (reliability of the polynomial approximation $R^2 = 0.881$). We suppose that this is due to the desire of the astronauts to clearly plan their activities, proactively proposing to MCC more efficient ways to use the crew's time. Thus, the study made it possible to identify the third and fourth quarters of the flight as problematic. However, not all identified differences are reliable and require further study.

### *3.3. Study of Astronauts' Communicative Styles*

Analysis of the three communication styles structure on days with different workloads showed that these structures remain unchanged; only the volume of communication differs (Figures 4–6). At the same time, each style has its own stable characteristics.

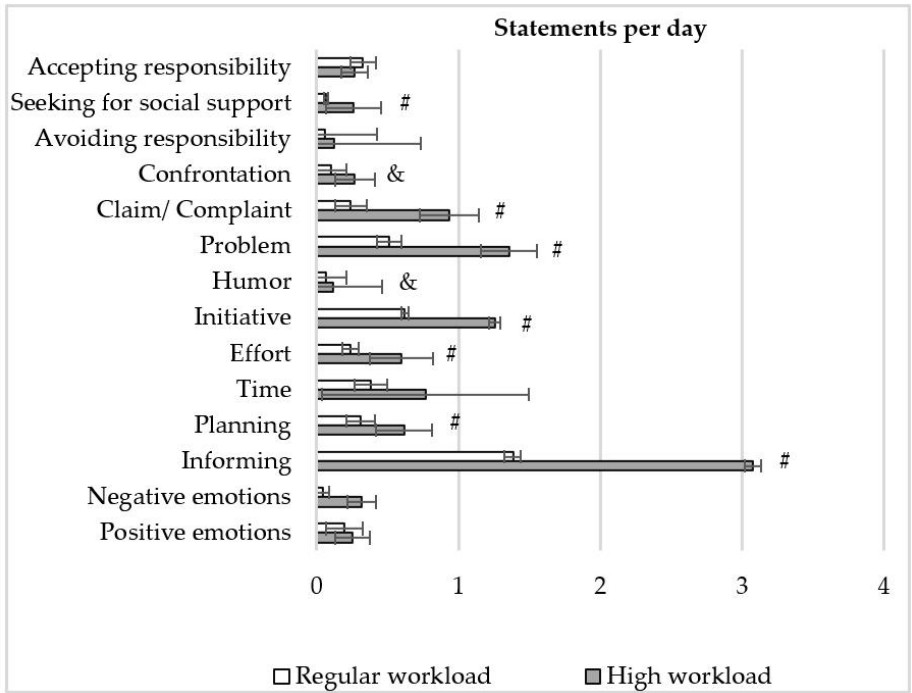

**Figure 4.** Distribution of statements within the "computing" communicative style by content analysis categories under regular and intensive workload (#—*p* < 0.05 and &—0.05 < *p* < 0.1).

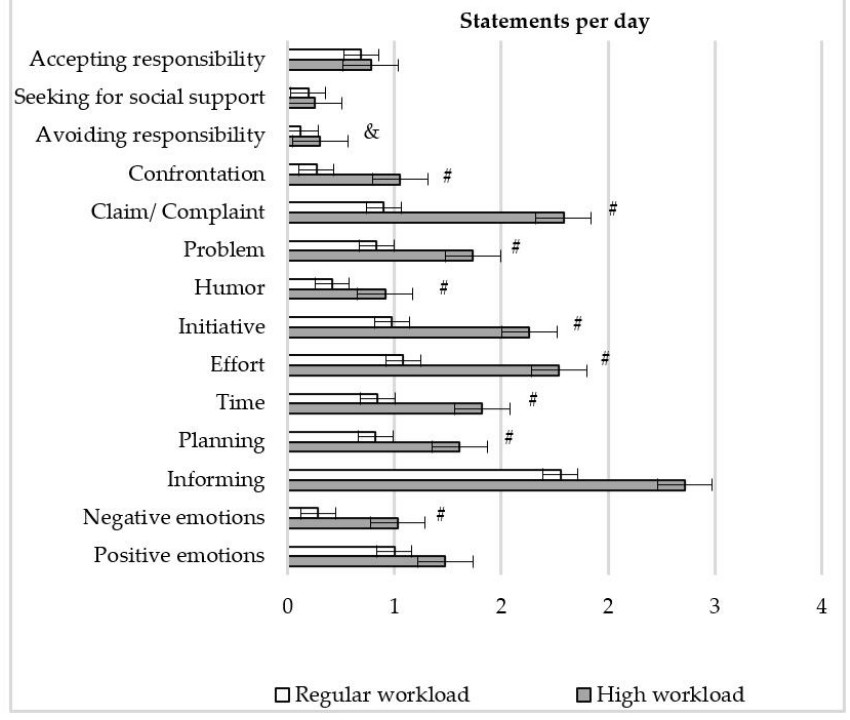

**Figure 5.** Distribution of statements within the "blaming" communicative style by content analysis categories under regular and intensive workload (#—*p* < 0.05 and &—0.05 < *p* < 0.1).

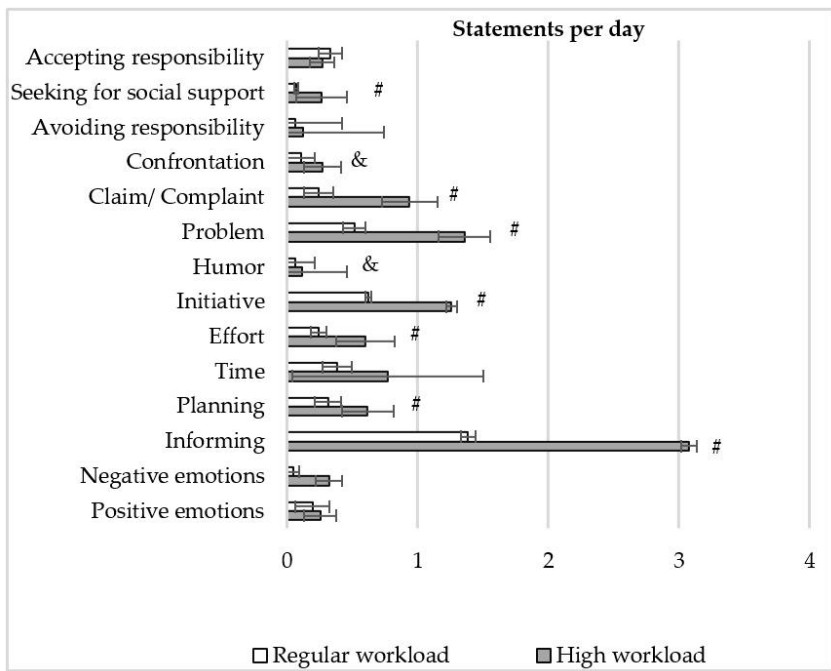

**Figure 6.** Distribution of statements within the "placating" communicative style by content analysis categories under regular and intensive workload (#—$p < 0.05$ and &—$0.05 < p < 0.1$).

The "computing" style manifested itself in regular reports to the Earth about what is happening onboard (*Informing*), combined with clarifying questions to specialists before making decisions. The solution of problems was carried out via their awareness and unambiguous "understanding" (this word is the most frequent verbal manifestation of this style, its semantic marker). *Informing* statements prevailed in speech; subjects showed a readiness to follow the plan (*Subordination*) while being *Initiative*, questioning the rationality of *Time* use, and constantly *Planning*. In problem situations, agreement or disagreement with the position of the MCC was rationalized; explanations were made in an emotionless manner. Three out of eight astronauts with a "computing" style had minimum manifestations of *Confrontation* coping (Figure 4).

The second most frequent communication style detected was "blaming." Distinctive features of this style are the intention to take control over problems by finding someone else responsible and proposing their solutions (Figure 5). Four subjects were included in this group. In routine communication, astronauts who expressed the "blaming" style, after a quick analysis of the problem (made with a certain irony), made counterproposals and expressed their *Initiatives* for correcting schedules (*Time* category). In this group, although the overall communication volume and *Informing* statements were higher than in the "computing" group, *Confrontation, Refusals*, and *Mistrust* were also present. In the worst cases, "blaming" communication led to emotional *Confrontation* and the expression of *Mistrust* in the competence of the interlocutor. Along with *Confrontation*, we found irony and sarcasm typical for blaming. This reaction is similar to what was described by several authors (e.g., [22]) as emotional transfer—a form of psychological defense allowing an astronaut to stabilize his psychological state via the draining of negative emotions accumulated during a long-term flight [6].

The "placating" style [14], noted in three subjects, was more common among young astronauts who made their first or second flight. The members of this group communicated with the MCC more than their experienced colleagues and more often informed specialists about what was happening onboard, seeking to obtain approval of what they were doing (*Subordination* and *Seek for support* categories). Thus, the overall communication volume of the "placaters" was the largest. Subjects with the "placating" style also experienced a

lack of *Time* for flight task completion more frequently (Figure 6). The verbal markers of "placating" that we detected were the words "help" and mentions of "lack of time."

Summing up these results, astronauts communicate with MCC specialists effectively and within the framework of a stable personal-inherent communication style, using a specific set of coping strategies.

*3.4. Emotional Transfer Phenomenon*

Our studies of interpersonal interaction within the crew in orbit confirmed the presence of specific phenomena of interpersonal tension among astronauts. The emergence of a problematic situation in orbit usually leads to an increase in psycho-emotional stress among astronauts during flight. One of the manifestations is the phenomenon of "emotional transfer", that is, the directing outwardly of tension and other unpleasant emotions from a person in an isolated group to a person outside the group who is a safe target; the displacement of negative emotions (anger and irritation) caused by activities and communication within the crew to external, safer interlocutors, in particular to MCC operators [23].

As we have already indicated above, the emotional transfer phenomenon manifests itself in situations of increased workload, when an increase in psycho-emotional stress causes an increase in statements related to stress coping. During these days, with a high workload, the number of statements reflecting conflict tension (*Confrontation and Claims/Demands/Requests*) as well as the emotional connotation (mostly negative) of the astronauts' messages significantly increases (Table 2). The transfer phenomenon was observed most clearly among astronauts with a "blaming" dominant communicative style (Figure 3) and to a lesser extent among those using "placating" and "computing" (Figures 2 and 4).

The transfer phenomenon manifested itself most clearly during the longest annual flight. That is, during expeditions ISS-43–56, the "drainage" of negative emotions via communication with the MCC ("emotional transfer" according to N. Kanas [23]) of negative experiences experienced during interaction with ground services was especially pronounced. In some cases, ineffective, from the astronauts' point of view, use of their time led to the appearance of counterproposals with a negative emotional connotation (category "Confrontation", reliability of the polynomial approximation $R^2 = 0.837$). These data confirm the results of American colleagues—responsible executors of Journals and Reaction Self-Test experiments, who obtained similar results in previous studies [7,17] and consider the "third-quarter phenomenon" to be a negative phenomenon requiring psychotherapeutic correction.

## 4. Discussion

In tense operating conditions, astronauts mobilize their psychological resources and apply their usual models of responding to problematic situations and resolving them in the context of interaction in communication with the MCC. Astronauts assess and cope with stressful situations of space flight in accordance with their personal characteristics, which determine the choice of coping strategies within a certain style of communicative behavior and depending on the communication functions prevailing in this style (exchange of information, exchange of emotions, and social regulation [6,10]). Different types and sets of coping strategies, however, have different effectiveness in remote joint activities. Among external factors, a significant influence on the astronauts' choice of coping strategies may be exerted not only by the nature of the problem situation itself but also by the MCC's communication specifics during joint problem solving. Thus, monitoring the manifestations of stress response in space flight using content analysis of communication makes it possible to access both the astronauts' psychological state dynamics and the effectiveness of interaction between crew and ground services.

The intensity of work and rest regimes significantly influences the volume and structure of communication between astronauts and the MCC, including the intensity of manifestation of characteristic coping strategies—usually aimed at instrumental problem solving

in case of increased workload (such as *Planning* and *Initiative*). At the same time, the emotional self-regulation of astronauts basically remains quite stable. Thus, during difficult and psychologically stressful situations and periods, astronauts, as a rule, tend to use effective communicative coping strategies.

Of interest are the phenomena of periodization that we may access using dynamics of the psychological state of astronauts during a long-term expedition that may be reflected in the communication volume and structure. The "third-quarter effect", previously discovered in long-term polar and submarine expeditions [24,25], may be observed in long-term (one year or more) space flights, being reflected in the intensification of the stress-coping strategies manifestations in astronauts' communication with the MCC. We believe that in longer and autonomous interplanetary flights, the phenomena of periodization would be more pronounced than in standard (six-month) low-orbit expeditions [26]. Based on the available data, we may assume that the appearance of periodization phenomena is influenced by the distribution of workload intensity; if it is sufficiently uniform and the workload is sufficiently (but not excessively) high, periodization is less pronounced.

The communication styles approach, based on V. Satir's classification [14,15], allows us to simplify and formalize the crew communication analysis by grouping individual characteristics of communication. Astronauts' communicative styles determine the structure of their communication with the MCC, more specifically, the prevalence of certain communicative functions and coping strategies manifestations. An astronaut's communicative style may depend on the experience. Astronauts making their first flights may tend to positively establish themselves despite a certain lack of experience and knowledge: these astronauts tend to use the "placating" style. But experienced astronauts frequently use the "blaming" style, actively criticizing the MCC and specialists for insufficient support, and imposing responsibility for unjustified expectations regarding the effectiveness of interaction.

The crew's manifestations of confrontation and negative emotions transferred toward the MCC may be associated with the excessively controlling, paternalistic communication style of the MCC, which pays no attention to crewmembers' subjective space flight experience and limits the astronauts' independence. Previously, both in model experiments and in space flights, there was evidence that this may be one of the drivers for the development of the "Us versus Them" phenomenon and may stimulate conflict tension between the crew and the MCC [27,28]. The traditional, directive, hierarchical, and "tutoring" structure of communication between the MCC and the crew, implying "parent–child" transactions, may turn out to be inadequate in ultra-long interplanetary flights. In ground confinement experiments that simulate ultra-long-term flights, there is an increase in crew autonomy, which is considered by a number of researchers to be useful in the context of promising research expeditions [29]. We believe that changes in the MCC's communication style, such as increasing expressions of trust in the crew and providing astronauts with greater freedom in planning activities, would reduce the likelihood of intergroup conflict tension and astronaut protests against excessive control.

The effectiveness of joint (in the "crew-MCC" circuit) problem solving largely depends not only on astronauts' communication styles but also on MCC's communication style and its adaptability (flexibility to adjust to an astronaut's individual communication style). One of the possible continuations of our study might be a content analysis of MCC personnel communications and a search for possible correlations between communication styles used by the two communication sides (both MCC personnel and crewmembers). At the same time, the MCC communication style, which is adequate for the objective situation and to a crewmember's individual characteristics, may serve as psychological support in the stressful conditions of space flight by providing the astronaut with necessary and appropriate assistance [30]. So, it is important to take into account both the specifics of the situation (its tension) and the communicative and behavioral style of the given astronaut. Reducing emotional tension, at the same time, does not only help to reduce the unfavorable psychological states among crewmembers, but also to increase the effectiveness of joint activities by improving relationships and increasing trust.

## 5. Conclusions

1. The intensity of work and rest regimes (workload intensity) significantly affect the volume and structure of astronauts' communication with the MCC. During difficult and psychologically stressful situations and periods, astronauts usually resort to effective behavioral strategies in order to cope with them.

2. The typology of communicative styles based on the classification of V. Satir makes it possible to simplify and formalize the approach to the analysis of crew communication by grouping the individual characteristics of astronauts' communication (such as manifestations of prevalent coping strategies). The style of the astronauts' communication with the MCC may depend not only on their personality traits but also—and even more so—on their experience.

3. The manifestations of confrontation on the part of the crew and the transfer (displacement) of their negative emotions toward the MCC may be associated with the excessively controlling communicative style of the Control Center. The effectiveness of joint (in the crew-MCC circuit) problem solving largely depends not only on the astronauts' communication styles but also on the communication style of the MCC and its adaptability.

4. The "third-quarter phenomenon", previously discovered in long-term polar expeditions, is mostly uncharacteristic for standard (6 months) space flights but can be registered in prolonged (1 year or longer) missions, being reflected in the manifestations of stress-coping strategies in the astronauts' interaction with the MCC. It can also be assumed that the occurrence of periodization phenomena is influenced by the distribution of workload intensity.

**Author Contributions:** Conceptualization, D.S., N.S. and A.Y.; Methodology, D.S., N.S. and A.Y.; Software, D.S., N.S. and A.Y.; Validation, D.S., N.S. and A.Y.; Formal Analysis, D.S., N.S. and A.Y.; Investigation, D.S., N.S. and A.Y.; Resources, D.S., N.S. and A.Y.; Data Curation, D.S., N.S. and A.Y.; Writing – Original Draft Preparation, D.S., N.S. and A.Y.; Writing – Review & Editing, D.S., N.S. and A.Y.; Visualization, D.S., N.S. and A.Y.; Supervision, D.S., N.S. and A.Y.; Project Administration, D.S., N.S. and A.Y.; Funding Acquisition, D.S. All authors have read and agreed to the published version of the manuscript.

**Funding:** This study was supported by the Russian Academy of Sciences FMFR-2024-0034.

**Institutional Review Board Statement:** All subjects gave their informed consent for inclusion before they participated in the study. The study was conducted in accordance with the Declaration of Helsinki, and the protocol was approved by IBMP Biomedicine Ethics Committee, Protocol number 303 and Human Research Multilateral Review Board, Protocol number 14-006.

**Data Availability Statement:** The data presented in this study are available upon request from the corresponding author. The data are not publicly available due to privacy restrictions.

**Acknowledgments:** The authors are deeply grateful to Nick Kanas for his valuable analytical approach to experimental data. We also express gratitude to our colleagues who have made significant contributions at various stages of this multi-year project: Angelina Chekalina, Polina Kuznetsova, and Alexandra Savinkina.

**Conflicts of Interest:** The authors declare no conflicts of interest.

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
