# Peer review of "The Communicative Behavior of Russian Cosmonauts: “Content” Space Experiment Result Generalization"

_aerospace, doi:10.3390/aerospace11020136_

Round 1

Reviewer 1 Report

Comments and Suggestions for Authors

The authors Sved et al. present a manuscript showing the results of their "Content" study wherein they analyzed communication styles of cosmonauts in their interaction with mission control. The investigated the semantic content of statements made by consomauts and analyzed them in the context of mission duration, character type and previous experience (missions flown before). This study is highly interesting as it might have major consequences for the future of space flight and might lead to the development of countermeasures. However, this manuscript needs to be better structured, especially the introduction needs to be rewritten. Moreover, the figures are not labelled correctly and sometimes there is no coherence between tables and the text.

In detail my comments are:

- Introduction:

1) The text is quite unstructured and is missing a clear introduction to the field of content analysis. Also, I recommend to introduce the reader to the third quarter phenomenon.

2) Some paragraphs that appear later in the method section appear much better suited for the introduction such as lines 97 to 131. Moreover, the expressions " in our opinion", "we believe" are better placed in the discussion section.

- Methods:

1) Accordingly, paragraph 2.4 contains a lot of information that is too long and unneccessary for the methods section and can be placed in the introduction or discussion. Please only describe what is needed to know in order to repeat the experiment or use the method in another study.

2) lines 170 to 182. You reference the works of V. Satir and reference five styles of communication in a closed loop. Later on you only mention 3 and then you only explain blaiming and computing. Please at least mention all five once and then also explain placating as it plays an important role in your results. This section could also be part of the introduction.

- Results:

1) Is the definition of clusters with high or standard workload a result from your study it is rather what is known and what you have defined as  workload intensity? Since docking/ EVAs are generally considered high workload phases I am assuming that you relied on what is known and this should then go into the methods section. The first figure is a result which should of course remain in the results section.

2) figure 1: please revise figure caption and explain what you are showing (Median, mean?)

3) figures in general: You show two times figures labelled with 2 and 3, respectively. Figure numbers need to be changed and also correctly referenced in the text.

4) Tables 3 and 4 are not referenced in the text.

- Discussion:

Please elaborate whether it might make sense to analyze communication styles of mission control personnel in the future and to monitor the way this daily interaction plays out. I am assuming that some of the communication you have observed might have been (unintentionally) influenced by ground personnel. Or was this standardized in your study?

Comments on the Quality of English Language

A minor spell check is suggested. There are minor things that need correction (e.g. first sentence of the discussion: "cosmonauts astronauts s")

Author Response

First, thank you for your thorough review which allowed us to enhance the quality of our manuscript.

We made changes to the manuscript in accordance with your suggestions. The corresponding edits are marked using green text coloring.

- Introduction:

Most part of the text considering the theoretical background and previous studies (including our own) was moved from the Methods section to the Introduction.

- Methods:

1) See above

2) Changes were made

- Results:

1) Both Results and Methods sections were edited accordingly

2-4) Figure labels are corrected, and we decided to remove Tables 3-4

- Discussion:

Corrections were made to the last paragraph of the Discussion (lines 528-531)

Some other minor corrections were also made to the text

Reviewer 2 Report

Comments and Suggestions for Authors

General comment.

This is a quite interesting paper, both from an informative perspective (it contains an historical overview of the matter, rather well presented) and especially for its operative value. It in fact includes a series of strategic points dealing with intra-operator communication, most of which may result helpful in the course of space missions. Those flight indications may well be exploited in the delicate phase of planning of a flight, particularly in the case of long-duration space missions.

A very useful part of the present contribution, therefore, is represented by the four final Conclusions. In particular, the last one provides an original and highly appreciable one.

Finally, the narrative-style of the text, while making it easily readable also for a general readership, at the same time repeatedly uses some excessively conversational terminology. This may confound the reader. Some requests to better define some terms or expressions are reported below.

Specific points

Introduction, page 1

Lines 28-29

more crucial as ultra-long-term (one year or more) flights, including interplanetary flights, are becoming a close perspective. We are aiming to show these communication peculiar

More than one year duration: which range of time?

Line 39

to maintain efficient data transmission and to maintain the optimal mental status 

“Optimal mental status” does need a better definition.

Page 2, 

Lines 78-80

The studies were conducted within the frame of “Content” space experiment involving Russian ISS crewmembers [1]. The experiment was dedicated to psycholinguistic analysis of crew-MCC communication.

Some more info on the “Content” experiments seems necessary.

Page 3,

 Lines 117-118

combined with an increase in emotionally charged statements

“Emotionally charged statement” does need a better definition.

Line 140

Ambivalent statements

“ambivalent statement” does need a more precise definition.

Page 4

Line 144 

spoken, but the statement: a fully expressed idea [4].

“Fully expressed idea” does need a better definition.

Lines 159-160

to react to stress and threats to their self-esteem

“Threat to a self-esteem” does need a better definition.

Page 5

Line 199

For our study, we chose the V. Satir’s classical communication model (1972) 

A paragraph should explain such a “classical” communication model. This will help understanding by a nonspecialized audience.

Page 9

Lines 319-320

We analyzed adaptation dynamics to space flight conditions in astronauts through the volume and structure of communication content, but obtained no clear results. 

Possibly, some results (even not reaching full statistical significance) may however be operatively relevant and potentially helpful (see also General comment). Such a negative statement could be lowered. (I however leave to the Authors the final decision about my present suggestion).

Page 12

Lines 397-398

included in this group. In routine communication, astronauts expressing “blaming” style, after a quick analysis of the problem (made with a certain irony),

Excessively conversational style?

Page 14,

Lines 434-437

One of the manifestations is the phenomenon of “emotional transfer”, that is, the displacement of negative emotions (anger, irritation) caused by activities and communication within the crew to external, safer interlocutors, in particular to MCC operators [25].

This is a central points possibly should be expanded, eg in terms of variability in crew vs MCC operators interactions.

Page 15

Lines 503-506

Crew’s manifestations of confrontation and negative emotions transfer towards the MCC may be associated with the excessively controlling communication style of the MCC, which makes no attention to crewmembers’ subjective space flight experience and limits the astronauts’ independence.

“Excessively controlling style” does need a better definition.

Lines 519-520

communication style and its adaptability

A better definition of adaptability seems a crucial issue, eg in terms of diversified coping responses

Author Response

Thank you for your thorough and helpful review!

We made corrections to the manuscript in accordance with all your comments. The corresponding edits are marked with red text color.

Due to some extensive edits in response to other reviewer, parts of the text were moved, please see the edits in the new positions:

Line 39 - now 41

Lines 78-80 - now 165-168

Lines 117-118 - now 97-98

Line 140 - now 187

Line 144 - now 191

Lines 159-160 - now 112-113

Line 199 - now 137

Communication styles (according to V. Satir) are now described in the Introduction (lines 124-136)

Lines 434-437 - now 440-445

Line 503 - now 512

Lines 519-520 - now 527-528